

# Cuproptosis related ceRNA axis AC008083.2/miR-142-3p promotes the malignant progression of nasopharyngeal carcinoma through STRN3

Dandan Feng, Xiaoping Wu, Genping Li, Junhui Yang, Jianguo Jiang, Shunan Liu and Jichuan Chen

Department of Otolaryngology Head and Neck Surgery, Army Special Medical Center (Daping Hospital), Army Medical University, Chongqing, Chongqing, China

## ABSTRACT

**Background**. CeRNA axis is an important way to regulate the occurrence and development of Nasopharyngeal carcinoma (NPC). Although the research on inducing cuproptosis of tumor cells is in the early stage of clinical practice, its mechanism of action is still of great significance for tumor treatment, including NPC. However, the regulation mechanism of cuproptosis in NPC by ceRNA network remains unclear.
**Methods**. The ceRNA network related to the survival of nasopharyngeal carcinoma related genes was constructed by bioinformatics. Dual-luciferase reporter assay and other experiments were used to prove the conclusion.
**Results**. Our findings indicate that the AC008083.2/miR-142-3p axis drives STRN3 to promote the malignant progression of NPC. By performing enrichment analysis and phenotypic assays, we demonstrated that the changes in the expressions of AC008083.2/miR-142-3p/NPC can affect the proliferation of NPC. Mechanistically, luciferase reporter gene assays suggested that AC008083.2 acts as a ceRNA of miR-142-3p to regulate the content of STRN3. Furthermore, the regulations of STRN3 and the malignant progression of NPC by AC008083.2 depends on miR-142-3p to some extent.
**Conclusions**. Our study reveals an innovative ceRNA regulatory network in NPC, which can be considered a new potential target for diagnosing and treating NPC.

## INTRODUCTION

Nasopharyngeal carcinoma (NPC) is a carcinoma originating in the nasopharynx. The disease has a poor prognosis and high malignancy (*Chen et al., 2019*). Early-stage NPC generally has a 60–75% five-year survival rate, but advanced-stage NPC has a much lower five-year survival rate of less than 40%, highlighting the challenges of treating cancers that have spread extensively at the time of diagnosis (*Wong et al., 2019*). Unsatisfactorily, NPC often goes undetected until a later stage, resulting in delayed diagnosis and limited treatment options (*Davidson et al., 2009*). In addition, it tends to spread early to regional lymph nodes and distant organs, contributing to its aggressive nature (*Pisani et al., 2020*). Regarding sedation methods for NPC treatment, although general anesthesia is commonly

Corresponding author
Jichuan Chen, chenjcmd-dpyy@163.com

used to ensure patient comfort during invasive procedures, there are limitations to this approach, including potential risks associated with anesthesia administration and the need for specialized facilities and personnel (*Wang et al., 2022c*). NPC typically involves a multimodal approach combining radiation therapy, chemotherapy, and sometimes surgery (*Anderson et al., 2021*). Radiation therapy has greater efficacy on NPC cells with high radiosensitivity (*Chen et al., 2021*). However, treatment limitations arise from the proximity of critical structures in the head and neck region, leading to significant side effects and complications. Therefore, new methods are needed to diagnose and treat NPC.

MicroRNAs (miRNAs) are small non-coding RNA molecules that can affect gene expression by directly binding to messenger RNA (mRNA), leading to degradation or translational repression (*Chen, Xu & Liu, 2019*; *Fernandes et al., 2019*). During the development of tumors, miRNAs are involved in key cellular processes such as cell proliferation, angiogenesis, apoptosis and metastasis (*Asl et al., 2021*). In NPC, specific miRNAs can act as oncogenes or tumor suppressor genes to affect various signaling pathways and target genes involved in NPC pathogenesis (*Lao, Quang & Le, 2021*). Understanding the miRNA-mediated mechanisms in NPC offers insights into its diagnosis, prognosis, and potential therapeutic targets. For example, MiR-18a and miR-17, from the miR-17-92 cluster, are often overexpressed, promoting proliferation and reducing apoptosis by targeting genes such as PTEN (*Zhao et al., 2022*). In contrast, miR-34a, a downregulated tumor suppressor, targets oncogenes and aids in apoptosis, indicating its potential as a therapeutic target (*Fu et al., 2023*). MiR-142-3p is a microRNA that plays a regulatory role in tumorigenesis (*Lawson et al., 2019*; *Troschel et al., 2018*). Biomarkers based on miR-142-3p have gradually been used for predicting the prognosis of nasopharyngeal carcinoma patients (*Zhang et al., 2019*). Long non-coding RNAs (lncRNAs) are a class of RNA molecules with limited protein-coding potential (*Kopp, 2019*). LncRNAs are crucial in gene expression regulation, chromatin modification, transcription, and post-transcriptional processing (*He, Luo & Mo, 2019*). LncRNAs have been demonstrated to be involved in tumorigenesis, metastasis, and invasion in NPC (*Fan et al., 2019*). However, more researches are still needed to thoroughly clarify the functions of lncRNAs in NPC and other cancers. As a kind of RNA molecule, competitive endogenous RNAs (ceRNAs) modulate the activity of other RNA transcripts *via* engaging in competition for shared miRNAs (*Ala, 2020*). LncRNAs have been shown to function as ceRNAs, and they up-regulate the expression of target genes by competing for miRNA binding (*Wang et al., 2019*). Dysregulation of ceRNAs is associated with many diseases, including NPC (*Tan et al., 2021*).

Copper, as an elementary basic trace element, can widely participate in cellular processes, such as energy production, antioxidant defense, and signaling pathways (*Zwierełło et al., 2020*). However, an imbalance in copper levels can result in cytotoxic effects (*Wang et al., 2018*). The excess copper generates reactive oxygen species (ROS), causing oxidative damage to cellular components or even triggering cell death (*Ji et al., 2023*). Cuproptosis is a newly discovered mode of cell death triggered by the dysregulation of copper homeostasis (*Chen, Min & Wang, 2022*). This process has gained attention in cancer research due to its potential implications for tumor growth and treatment response (*Tong et al., 2022*). Cuproptosis has been shown to have both tumor-promoting and tumor-suppressing

effects (*Xie et al., 2022*). On the one hand, cuproptosis can contribute to tumor growth by inducing cell death in surrounding healthy cells, creating space for tumor expansion, and providing nutrients released from dying cells (*Chen et al., 2023*). On the other hand, cuproptosis can suppress tumors by eliminating cancer cells that have acquired copper overload due to genetic alterations or exposure to chemotherapy agents (*Huang et al., 2022*). Researchers have conducted extensive studies on cuproptosis-related genes and Jab1 in the tumor microenvironment of NPC (*Wang et al., 2022b*). Cuproptosis-related genes related to Jab1 affect the tumor microenvironment and pharmacological profile in NPC. This finding suggests that cuproptosis may contribute to the progression of NPC. Understanding the role of cuproptosis in NPC provides potential avenues for therapeutic interventions.

STRN3 (Striatin 3), also known as SG2NA (Striatin, Gametogenesis-2, Neural), is a protein-coding gene that is important in many cellular programs, including cell signaling, cell cycle regulation, and cellular differentiation (*Tanti et al., 2023*). STRN3 is a striatin family of scaffolding proteins involved in protein-protein interactions and signal transduction pathways (*Afza et al., 2022*). STRN3 is associated with cancer development and progression. Its dysregulation can affect various cellular processes contributing to tumor formation (*Sarmasti Emami, Zhang & Yang, 2020*). It also can promote cell proliferation, invasion, and metastasis in certain cancer types (*Migliavacca et al., 2022*). It interacts with other proteins involved in cell signaling pathways, such as protein phosphatase 2A (PP2A), and can modulate its activity, affecting cell growth and survival (*Baskaran & Velmurugan, 2018*). STRN3 shows potential as a significant biomarker and therapeutic target in NPC (*Bryant et al., 2021*). Compared with normal tissues, aberrant expression of STRN3 has been observed in NPC tissues (*Verbinnen et al., 2021*). STRN3 affects NPC progression by promoting cell proliferation and inhibiting apoptosis (*Bryant et al., 2021*). Additionally, the dysregulation of STRN3 may contribute to the activation of oncogenic pathways and the development of treatment resistance in NPC (*Migliavacca et al., 2022*). Furthermore, the relationship between STRN3 expression and clinical outcomes in NPC patients has been highlighted in a previous study (*Verbinnen et al., 2021*). High expression levels of STRN3 can result in poor prognosis and reduced overall survival (OS) rates (*Liu et al., 2022*). The expression levels of STRN3 may be a valuable prognostic indicator and a potential therapeutic target for NPC. However, the regulatory mechanism by which STRN3 receives ncRNAs in NPC has yet to be elucidated.

This article found that as a ceRNA for miR-142-3p, AC008083.2 was regarded to regulate the expression of the cuproptosis-related gene STRN3 in NPC. The regulatory mechanisms of AC008083.2, miR-142-3p, and STRN3 on the malignant development of NPC were identified. Subsequently, we predicted and verified that AC008083.2 binded to miR-142-3p and regulated STRN3 as its molecular sponge. Morever, the regulation of STRN3 expression and NPC malignant progression by AC008083.2 was related to miR-142-3p.

## METHODS

### Data collection

The 29 pairs of paired data from cancer and para-carcinoma samples of NPC patients were downloaded from The Cancer Genome Atlas (TCGA) database. The data was used for differential gene expression analysis of lncRNAs and miRNAs. The relevant data and clinical information of 548 NPC patients were also downloaded from the TCGA database. The data was used for univariate Cox analysis of lncRNAs and miRNAs. The data can be found at the following website: https://portal.gdc.cancer.gov/projects/TCGA-HNSC. The project ID is TCGA-HNSC and the study accession number is phs000178.

### Differential gene expression analysis

The differentially expressed miRNAs, mRNAs, and lncRNAs were analyzed and screened using the R package "limma" (3.52.4), which was ground on logFC $\geq 1$ and $P < 0.05$. The "ggplot2" (3.4.0) and "pheatmap" (1.0.12) packages created volcano maps and heat maps to visualize the exhibition of DEGs.

### Gene set enrichment analysis (GSEA)

The method was used to analyze the main enrichment pathways of differentially expressed genes. The larger the enrichment factor, the more significant the enrichment level of differentially expressed genes in that particular pathway. The R package "clusterProfiler" (4.4.4) performed the Gene Ontology of DEGs. The accumulation of genes in the matching pathways is considered significant when $P < 0.05$.

### Consistent cluster analysis

The mRNA expression matrix of NPC patients was analyzed by the "Consensus ClusterPlus" (1.60.0) package. The consensus cumulative distribution function, consensus matrix, and consensus heat map were performed to determine the optimal cluster number.

### Univariate Cox analysis

Univariate Cox analysis was used to screen prognostic differential genes ($P < 0.05$) and construct a prognostic risk model based on gene expression. It was used to assess the overall survival (OS) and survival state of patients in the database.

### Pearson's correlation coefficient

To assess the linkage between two variables, Pearson's correlation coefficient was utilized. A strong association between the variables is suggested by a coefficient nearing either 1 or −1. Conversely, a coefficient approaching 0 implies a lack of correlation between them.

### CCK8 assay

The CCK8 kit (C0038; Beyotime) was implemented to explore cell proliferation. The cells were inoculated in a 96-well plate with 1000 cells per well. The cells were cultured in a 37 °C incubator for 24 h, 48 h and 72 h. Then, 10 μL CCK-8 reagent was added to the cells for 2 h. Finally, the microplate reader to detect the absorbance values of each well at 450 nm.

## Cell culture

The C666-1 cell line, specific to nasopharyngeal cancer, was acquired from ATCC. RPMI-1640 medium, enriched with 10% fetal bovine serum, was used for cell cultivation. These cells were maintained in an environment of 37 °C and 5% $CO_2$.

## Clone formation assay

C666-1 single cells were plated in six-well plates, each well containing $1 \times 10^3$ cells. Following a 10-day incubation period, colonies became apparent. For fixation, polyformaldehyde was applied, and crystal violet was used for staining the cells.

## RNA extraction and qRT-PCR

TRIZOL reagent (15596026; Invitrogen, Waltham, MA, USA) was utilized to extract total RNA from C666-1 cells and the quality and quantity of RNA was assessed though agarose gel electrophoresis (1%, 140 V, 25 min) and NANODROP ONE$^c$ (Thermo scientfic, 701-058108, America). The Prime Script RT kit (RR047A; TaKaRa, Tokyo, Japan) facilitated cDNA synthesis with the volume of 20 μL and 1000 ng RNA included. The RNA's relative expression was determined by the ratio of transcript levels in the sample through SYBR Premix Ex Taq™ II (RR820A; TaKaRa, Tokyo, Japan), using $\beta$-actin as a normalization control. The primers of RT-qPCR employed were as follows: AC00808 3.2 forward, 5′-AGAGGGTAAGCATGCAAGGT-3′ and reverse, 5′-GGTCTTGAACTCCTGGGCTA-3′; has-miR-142 -3p, 5′-TGTAGTGTTT CCTACTTTATGGA-3′; STRN3 forward, 5′-TGGCACAGAATGGGCTGAAC-3′ and reverse, 5′-TCCAAGGCCCAGTACACTT-3′; $\beta$-actin forward: 5′-ATGTGGCCGA GGACTTTGATT-3′ and reverse, 5′-AGTGGGGTGGCTTTTAGGATG-3′. The PCR program ran at CFX96 Real-Time System (CFX96; Bio-Rad, Hercules, CA, USA) as follows: start with initial denaturation at 95 °C for 30 s; Then, a 40 cycles amplification program with denaturation at 95 °C for 5 s and annealing at 60 °C for 30 s was performed; The melting curve analysis was performed after the cycles ended. All the information was collected and the relative fold-change was calculated with the formula $2^{-\Delta\Delta CT}$.

## Dual-luciferase reporter assay

Amplify the wild-type (WT) and mutant (Mut) AC008083.2 3′ UTR regions by PCR, inserting them into a luciferase-expressing plasmid (pGL4.15). C666-1 cells were seeded into 24-well plates. After cell adhesion, the indicated luciferase reporter and renin luciferase reporter were transfected together for 36 h by Lipofectamine 3000 (L3000150; Invitrogen). Luciferase activity was analyzed by renilla and firefly Lucifase®Reporter assay system (E1910; Promega, Madison ,WI, USA).

## Transfection

Lipofectamine 3000 (L3000150; Invitrogen) can transfect the inhibitors, mimics, and ASOs into C666-1 cells. The 293T cells were transfected with shRNA plasmids and packaging plasmids. The C666-1 cells were infected with the virus solution. The sequences of shRNAs are shown below:

 shSTRN3-1:5′-GCCAGUUAACGUGGAAGCATT-3′;
 shSTRN3-2:5′-GGAGGAGGCAAGUCAUUUA-3′.

## Construction of the ceRNA network

Firstly, we predicted the gene expression profiles of mRNAs, lncRNAs, and miRNAs in the database, and then predicted the binding sites of miRNAs. This step was also a key step in the ceRNA construction. Finally, based on the predicted results, Cytoscape software was used to construct molecular networks including lncRNAs, mRNAs, and miRNAs. Cytoscape software could display various informations on the network and visualize regulatory patterns.

## Statistical analysis

Data were gathered from three independent experimental replicates. The LSD-t test was utilized for comparing two sets of data, while ANOVA was employed for the analysis of multiple groups. Statistical calculations were conducted using GraphPad Prism 8. A $p$-value less than 0.05 was deemed significant. Survival curves were plotted using the Kaplan–Meier method.

# RESULTS

## Screening of lncRNAs, miRNAs, and mRNAs related to the progression of NPC

Tumor markers are widely involved in various processes of tumor development, and the level of tumor markers will change expressively during tumor progression. However, the part of the ceRNA axis for NPC has not been elucidated. To clarify the internal mechanism, the 29 pairs of paired data from cancer and para-carcinoma samples of NPC patients was used for differential expression analysis of lncRNAs and miRNAs. The relevant data and clinical information of 548 NPC patients were used for univariate Cox analysis of lncRNAs and miRNAs. (Figs. 1A and 1B and Fig. S1A). Then, we took the intersection of the up-regulated genes from the differential expression analysis and the oncogenes from the univariate Cox analysis. Similarly, we took the intersection of the down-regulated genes from the differential expression analysis and the tumor suppressor genes from the univariate Cox analysis. After the intersections, up-regulated oncogenes and down-regulated tumor suppressor genes were obtained (Fig. 1C).

## Consensus cluster analysis of cuproptosis-related genes

Cuproptosis plays a critical role in NPC. Regulation of cuproptosis related genes shapes the tumor microenvironment and pharmacological characteristics in NPC. Our research conducted a clustering analysis on NPC patients based on the expression matrix of 13 genes associated with cuproptosis. And the patients were divided into groups with the expression characteristics of cuproptosis related genes. A total of 548 patients with NPC from TCGA were divided into the $k$ group ($k = 2$–9). When $k = 3$, the optimal data classification was obtained (Fig. 2A and Figs. S2A and S2B). The group 2 and the group 3 with the largest difference in survival rate were selected for comparison. Kaplan–Meier analysis of the differences in overall survival (OS) between the two clusters found that patients in group 3 had worse OS (Fig. 2B). Limma package was used for differential expression analysis between group 2 and group 3, and 248 up-regulated and 4556 down-regulated differentially

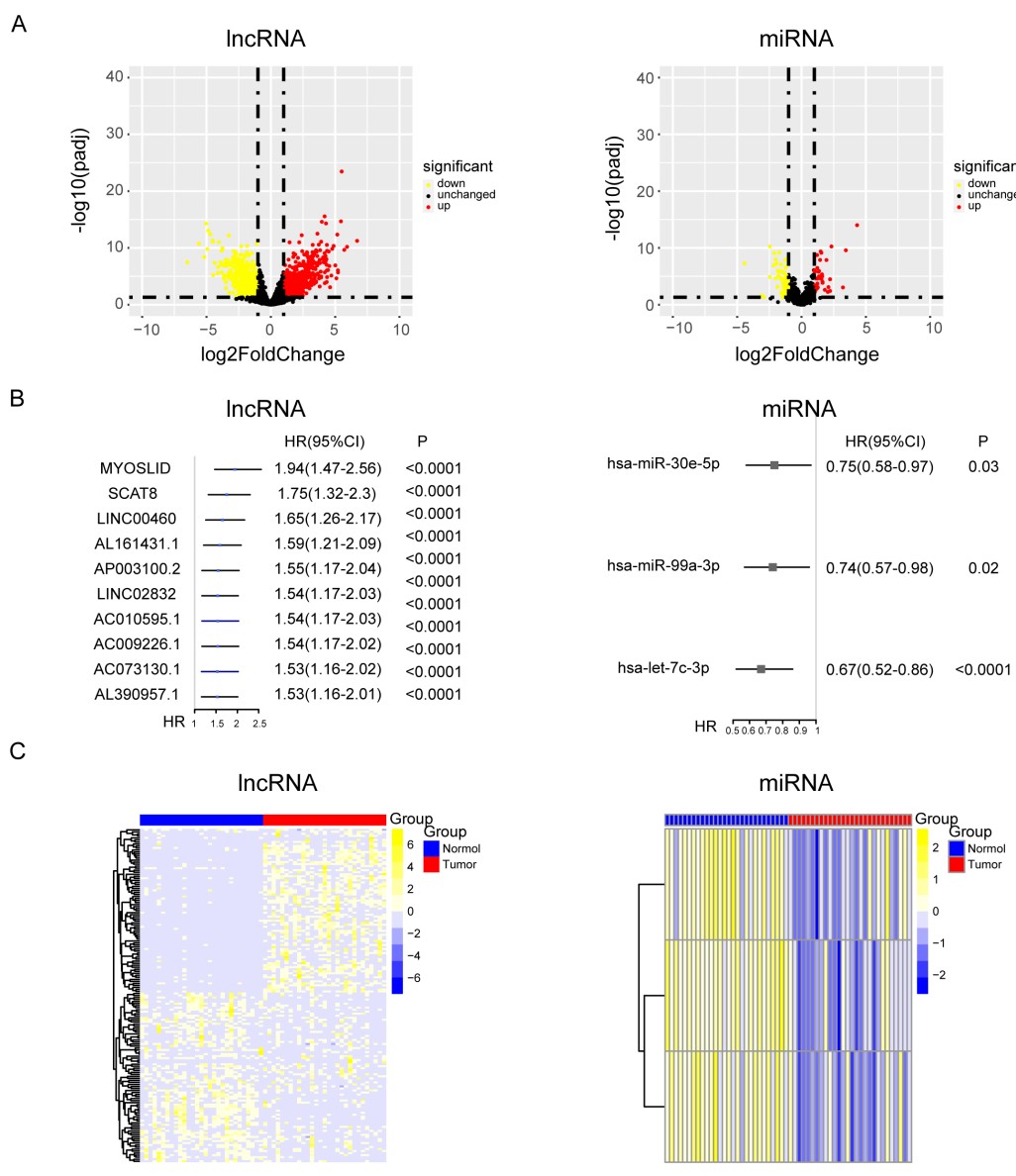

**Figure 1** **The results of univariate cox and differential expression analysis of lncRNAs and miRNAs from the TCGA-NPC database.** (A) Volcano plots of differential gene expression analysis of lncRNAs and miRNAs in cancer and para-carcinoma samples of NPC patients. (B) The top ten results of between univariate cox and differential expression analysis in lncRNAs and miRNAs. (C) The heat maps of lncRNAs and miRNAs in cancer and para-carcinoma samples of NPC patients.

expressed genes were observed (Figs. 2C and 2D and Fig. S2C). The ceRNA network was constructed by combining the mRNAs of NPC samples with the datas of cuproptosis cluster analysis and univariate Cox analysis (Fig. 2E and Fig. S2D).

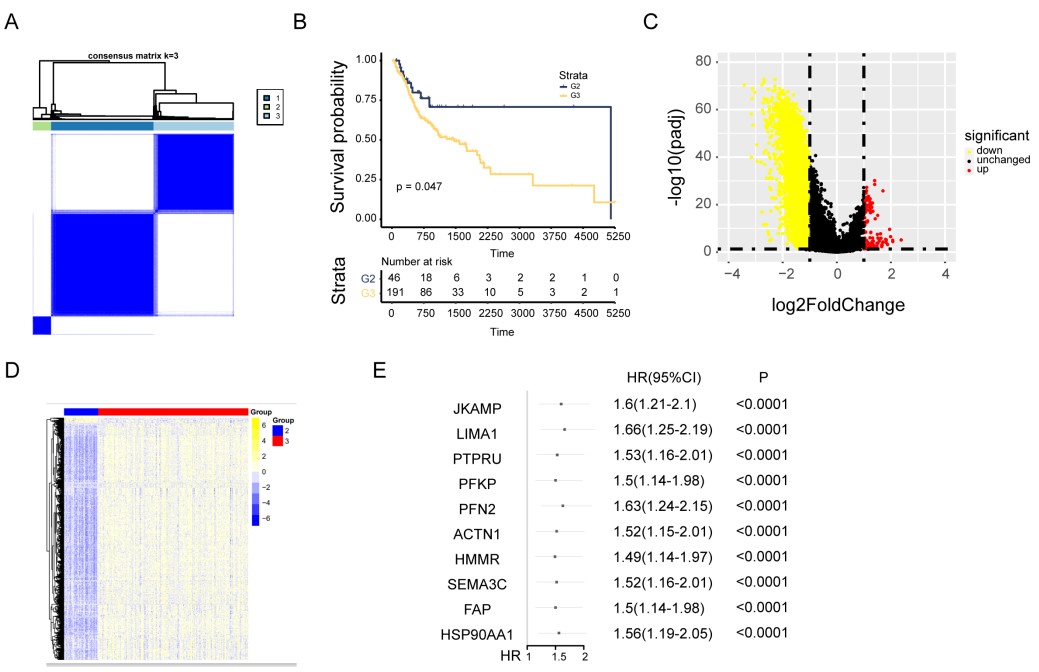

**Figure 2  Consensus cluster analysis of genes related to cuproptosis.** (A) Consensus clustering heatmap of genes associated with cuproptosis with $k = 3$. (B) Kaplan–Meier survival curve between cluster 2 and cluster 3. (C) Volcano map showed the differential genes of the two clusters. (D) The heat map of differential genes. (E) The top ten results of between univariate cox and differential expression analysis in mRNAs.

## AC008083.2/miR-142-3p/STRN3 may be an important ceRNA axis regulating NPC

Based on the above results, the mRNAs, lncRNAs, and microRNAs obtained after the intersection were used to construct the ceRNA network to regulate the development of NPC. The network contains three lncRNAs, one miRNA, two mRNAs, and six interaction axes (Figs. 3A–3B and Fig. S3). STRN3 is a cuproptosis related gene and plays an important role in tumors. We speculated that STRN3 might regulate the cuproptosis of NPC and promote its malignant progression. However, the relationship between MORF4L2 and tumors has not been clarified. Therefore, we focused on STRN3 for this article. To clarify the key lncRNAs regulating STRN3, differential expression analysis and survival analysis were conducted for AC008083.2, AC078820.1, and AL160408.4. The results of differential expression analysis showed that AC008083.2 and AL160408.4 were significantly overexpressed in NPC (Fig. 4A). Therefore, the lncRNAs may be closely related to the high expression of STRN3. Furthermore, the survival analysis and correlation analysis demonstrated that the lncRNA AC008083.2 was associated with the worse survival and the higher correlation of STRN3 (Fig. 4B and Fig. S4). For further screening results, AC008083.2, AC078820.1, and AL160408.4 were knocked down with ASO, and the alterations in nucleic acid levels of STRN3 were detected (Fig. 4C). The results confirmed that AC008083.2 could significantly

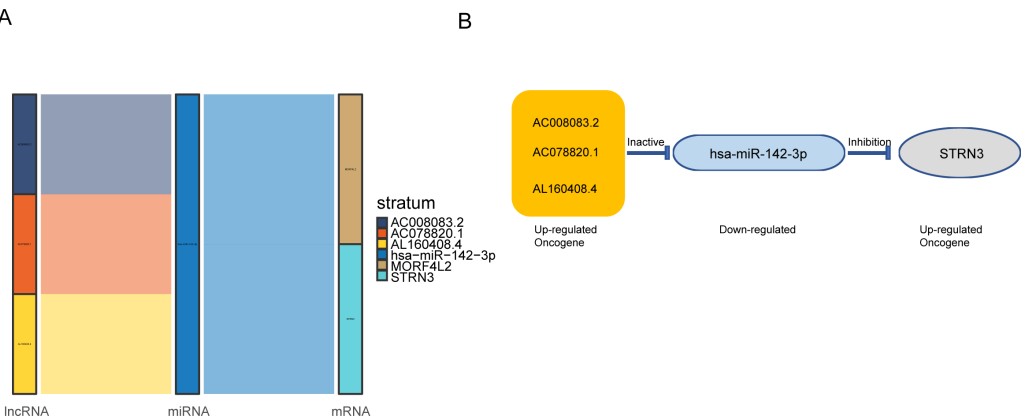

A

B

stratum
- AC008083.2
- AC078820.1
- AL160408.4
- hsa-miR-142-3p
- MORF4L2
- STRN3

lncRNA          miRNA          mRNA

**Figure 3   AC008083.2/miR-142-3p/STRN3 may be an important ceRNA axis in the regulation of nasopharyngeal carcinoma.** (A) Sankey diagram for the ceRNA network in Nasopharyngeal carcinoma. Each rectangle represents a gene, and the connection degree of each gene is visualized based on the size of the rectangle. (B) The ceRNA regulatory network.

regulate the expression of STRN3, while other lncRNAs were not significant enough. In summary, AC008083.2 may be an important lncRNA regulating STRN3.

## AC008083.2, miR-142-3p, and STRN3 regulate the malignant progression of NPC

Based on the above-predicted results, AC008083.2/miR-142-3p/STRN3 might be the ceRNA axis that regulated NPC progression. However, their role in the development of NPC has not been illustrated. Thus, based on the high and low expressions of the lncRNA AC008083.2, the NPC samples were classified into two groups for differential expression analysis and GSEA. The same analysis was conducted for miR-142-3p and STRN3. All above results revealed that the genes regulated by the three genes were largely enriched in the cell proliferation pathways (Fig. 5A and Fig. S5). To explore the effects of AC008083.2, miR-142-3p, and STRN3 on NPC cell proliferation, we conducted CCK8 and cell colony formation experiments. The results showed that when AC008083.2 and STRN3 were knocked down, cell proliferation was inhibited, while the inhibition of miR-142-3p expression promoted the proliferation of NPC cells. In summary, these three genes affected the regulation of C666-1 cell proliferation. The CCK8 and cell colony formation results also conformed to the results of differential expression analysis (Figs. 5B and 5C). These results indicated that three genes could regulate NPC progression.

## LncRNA AC008083.2 and miR-142-3p could regulate STRN3 in NPC

Next, we validated the correlations among the three genes. After knocking down AC008083.2 with ASO, the results of q-PCR and western blot indicated that the expression of STRN3 was decreased, while the expression of miR-142-3p was increased (Fig. 6A). Subsequently, after overexpression and repression of miR-142-3p using the corresponding mimic and inhibitor, we detected changes in the expression of AC008083.2 and STRN3. The results showed that AC008083.2 and STRN3 were negatively regulated by miR-142-3p at

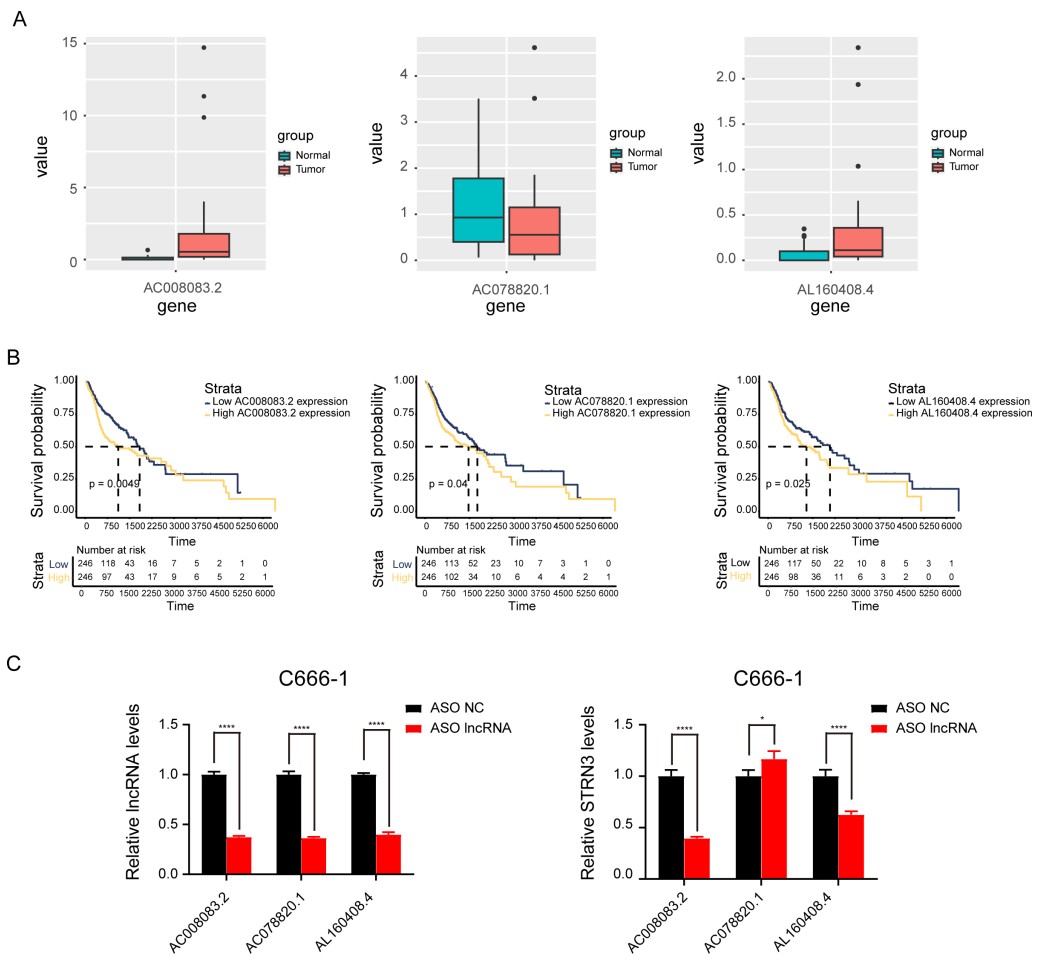

**Figure 4 AC008083.2 may regulate the RNA level of STRN3.** (A) The expression of pontential lncRNAs between tumor and normal tissues in nasopharyngeal carcinoma patients. The red box indicates the tumor tissue and the green box indicates the normal tissue. (B) The survival curves of pontential lncRNAs in nasopharyngeal carcinoma patients. Black lines indicate low expression of lncRNA and red lines indicate high expression of lncRNA. (C) STRN3 mRNA levels in C666-1 cells with three potential lncRNAs knockdown by ASO (ASO NC group: $n = 3$; ASO lncRNA: $n = 3$) (*, $P < 0.05$, ****, $P < 0.0001$).

RNA levels and protein levels (Figs. 6B and 6C). In order to determine whether AC0080832 was a ceRNA of miR-142-3p, we performed a complementary mutation at the predicted binding site of AC0080832 and miR-142-3p (Fig. 6D). In addition, we tested the effect of miR-142-3p on luciferase activity before and after the AC0080832 mutation. The luciferase reporter gene experiment showed that the luciferase activity of wild-type AC008083.2 was regulated by miR-142-3p, while mutant AC00808.2 was not regulated (Fig. 6E). All above results indicated the lncRNA AC008083.2 and miR-142-3p could regulate STRN3 in NPC tumors, which is in line with the trend of the ceRNA axis.

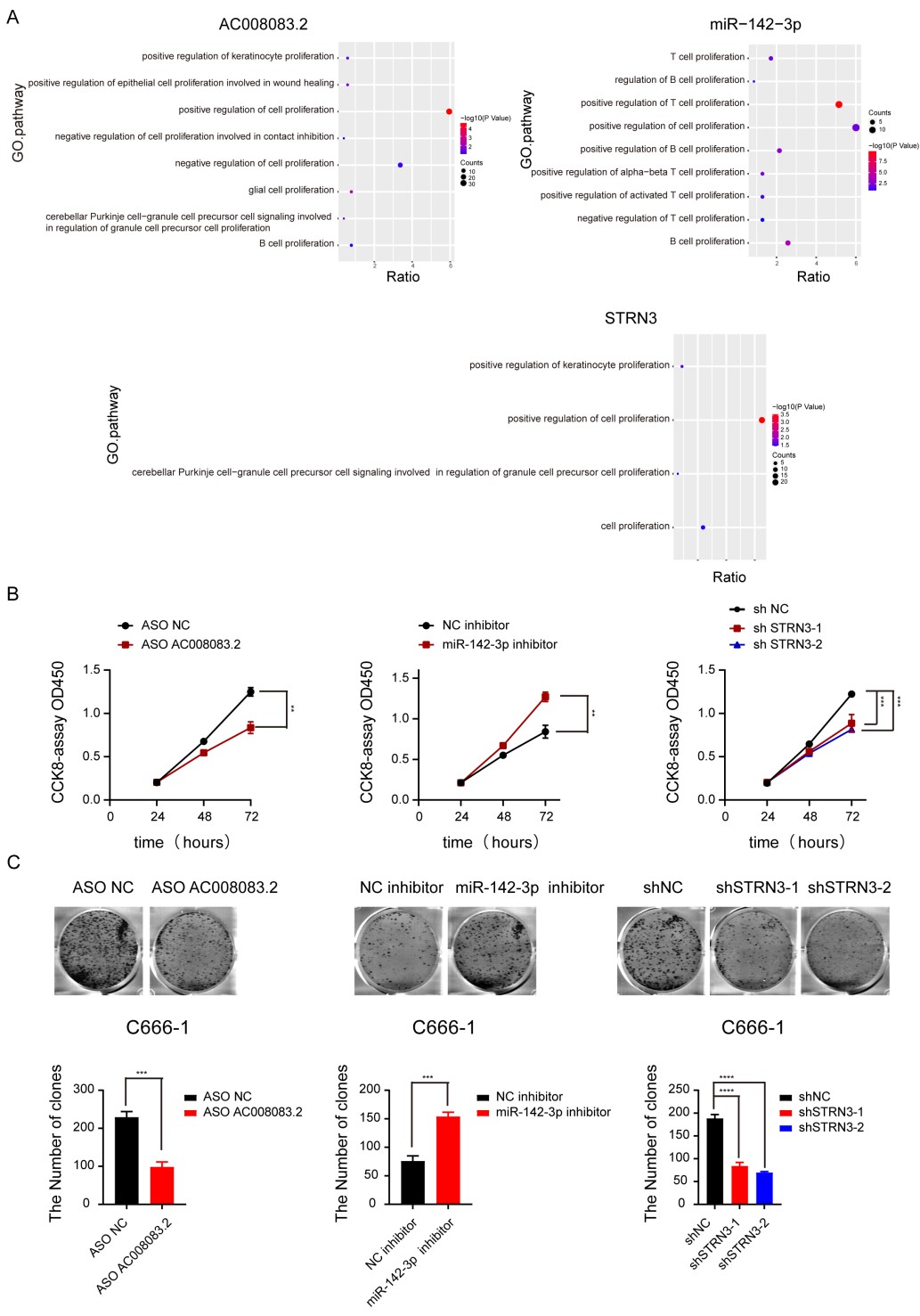

**Figure 5** **AC008083.2, miR-142-3p and STRN3 regulate the malignant progression of nasopharyngeal carcinoma.** (A) GSEA enrichment map of AC008083.2, (continued on next page...)

**Figure 5 (…continued)**
miR-142-3p and STRN3. (B) The CCK8 assays of C666-1 cells were performed when AC008083.2, miR-142-3p and STRN3 knockdown or inhibited (ASO NC group: $n = 3$; ASO lncRNA: $n = 3$; NC inhibitor group: $n = 3$; miR-142-3p inhibitor group: $n = 3$; shNC group: $n = 3$; shSTRN3-1 group: $n = 3$; shshSTRN3-2 group: $n = 3$) (**, $P < 0.01$, ***, $P < 0.001$). (C) The clone formation assays of C666-1 cells were performed when AC008083.2, miR-142-3p and STRN3 knockdown or inhibited (ASO NC group: $n = 3$; ASO lncRNA: $n = 3$; NC inhibitor group: $n = 3$; miR-142-3p inhibitor group: $n = 3$; shNC group: $n = 3$; shSTRN3-1 group: $n = 3$; shshSTRN3-2 group: $n = 3$) (***, $P < 0.001$, ****, $P < 0.0001$).

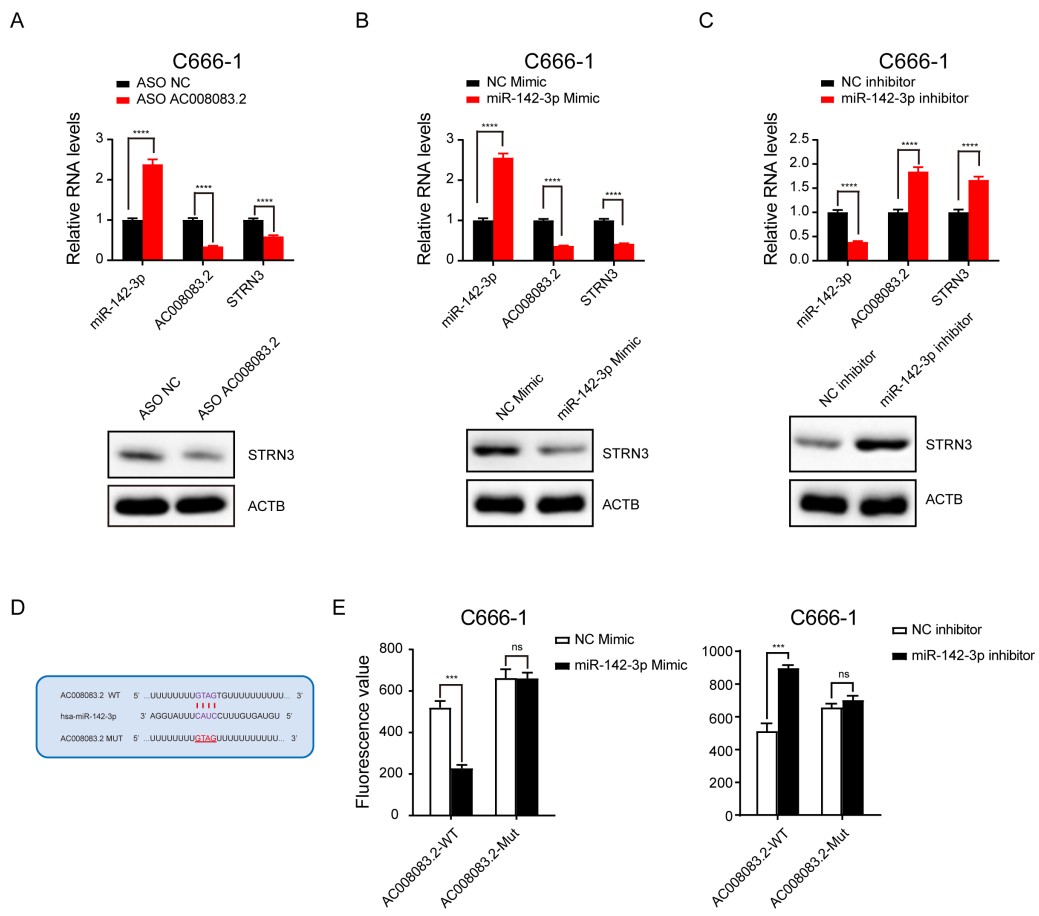

**Figure 6  LncRNA AC008083.2 and miR-142-3p could regulate STRN3 protein in nasopharyngeal carcinoma.** (A–C) RT-qPCR and western blot were used to detect the expression levels of AC008083.2, miR-142-3p and STRN3 after knocking down AC008083.2 with ASO or treating with miR-142-3p mimic or inhibitor for 48 h, respectively (ASO NC group: $n = 3$; ASO AC008083.2 group: $n = 3$; NC mimic group: $n = 3$; miR-142-3p mimic group $n = 3$; NC inhibitor group $n = 3$; miR-142-3p inhibitor group $n = 3$) (****, $P < 0.0001$).(D) Predicted binding site between miR-142-3p and AC008083.2. (E) Luciferase assays were performed to test the effect of miR-142-3p on wild-type or mutant AC008083.2 after treating with miR-142-3p mimic or miR-142-3p inhibitor (NC mimic group: $n = 3$; miR-142-3p mimic group $n = 3$; NC inhibitor group $n = 3$; miR-142-3p inhibitor group $n = 3$) (***, $P < 0.001$, ns, no statistical significance).

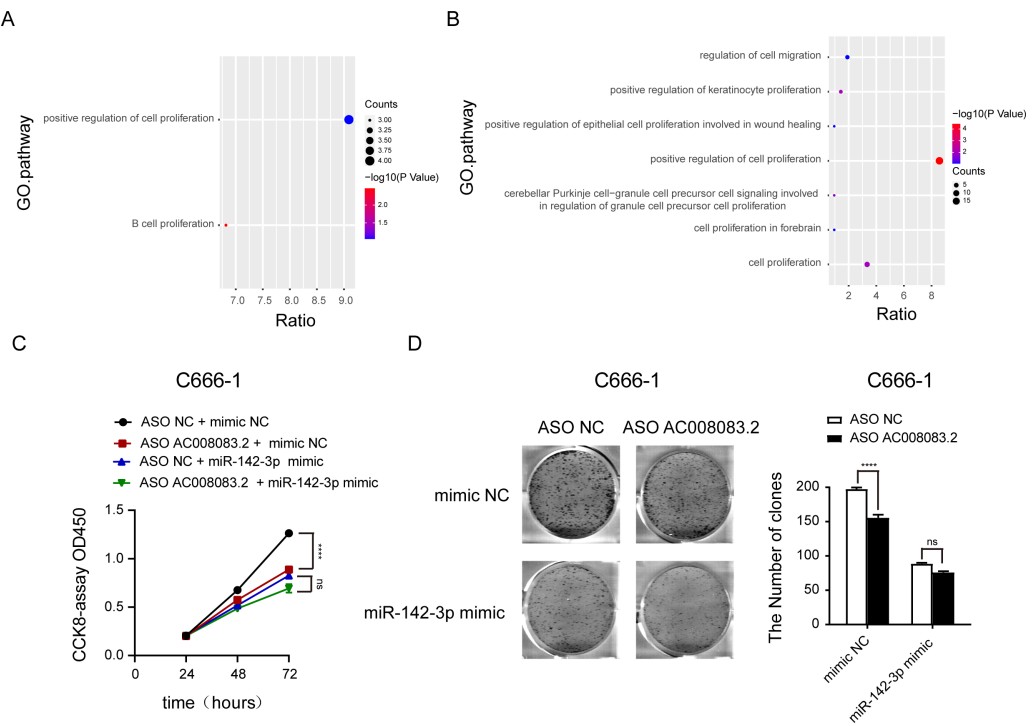

**Figure 7  AC008083.2 as a ceRNA of miR-142-3p regulates STRN3 protein to affect the malignant progression of nasopharyngeal carcinoma.** (A) Pathway enrichment of genes co-regulated by AC008083.2 and miR-142-3p. (B) Pathway enrichment of genes co-regulated by AC008083.2 and STRN3. (C)The CCK-8 results of C666-1 cells with knockdown AC008083.2 treated with NC mimic or miR-142-3p mimic (each group in each time, $n = 3$) (****, $P < 0.0001$, ns, no statistical significance). (D) The clone formation assays of C666-1 cells with knockdown AC008083.2 treated with NC mimic or miR-142-3p mimic ($n = 3$) (****, $P < 0.0001$, ns, no statistical significance).

## AC008083.2, as a ceRNA of miR-142-3p, regulates STRN3 to affect the malignant progression of NPC

Then, we labeled the abundance of genes regulated by AC008083.2 and miR-142-3p or AC008083.2 and STRN3 in the progression pathway of NPC. This important finding further emphasized that the enrichment of genes in the proliferation pathway was particularly meaningful (Figs. 7A and 7B, and Fig. S6). To verify this finding, we detected the cell proliferation in the control group and miR-142-3p mimic group with AC008083.2 knockdown. The results indicated that knocking down AC008083.2 could inhibit the proliferation of NPC cell C666-1 in the control group, while in the mimic group, there was no significant inhibitory effect (Figs. 7C and 7D). All above results suggested that AC008083.2, as a miR-142-3p ceRNA, regulated STRN3 and affected the development of NPC.

## DISCUSSION

NPC has a poor prognosis and high malignancy currently. Early-stage NPC (I–II) is predominantly treated with radiotherapy, known for side effects like dry mouth and

hearing loss. Advanced stages (III–IV) necessitate combined modalities like chemotherapy, which presents side effects like nausea and hair loss, and targeted therapies, which are more selective but have their own adverse effects. Treatment efficacy is generally favorable in early stages, but outcomes become more variable as the disease progresses, underscoring the importance of early detection and individualized therapeutic approaches. Non-coding RNAs, including lncRNAs and miRNAs, have shown their importance in NPC. The dysregulation of lncRNAs is associated with a range of diseases, including cancer. Furthermore, lncRNAs are involved in tumorigenesis, metastasis, and invasion in NPC. The lncRNA HOTAIR has been identified as a significant contributor to its development and progression (*Rajagopal et al., 2020*). Other lncRNAs, such as MALAT1, SNHG1, and PVT1, are also associated with NPC (*Wang, Wang & Fan, 2022*). MiRNAs have also been shown to be dysregulated in NPC and involved in tumorigenesis, metastasis, invasion, resistance to chemotherapy, and multidrug resistance. Recent studies have highlighted the potential of miRNAs as therapeutic targets for NPC. Specifically, miR-142-3p has been identified as a key player in cancer development. Therefore, lncRNAs and miRNAs have emerged as potential therapeutic targets for NPC. More studies are needed to fully understand the roles of lncRNAs and miRNAs in NPC and other cancers. LncRNAs and miRNAs regulate the development of NPC through ceRNA. For example, LINC02570 is a lncRNA that functions as a ceRNA to up-regulate the expression of a core gene involved in NPC proliferation (*Wang et al., 2022a*). Additionally, miR-490-3p is a miRNA regulated by the lncRNA HCG11 and plays a role in NPC progression by targeting the MAP3K9 gene (*Zheng et al., 2022*). CeRNA networks can also promote NPC proliferation by inhibiting the expression of the tumor suppressor PTEN (*Zhang et al., 2022*). These findings indicate that lncRNAs and miRNAs are crucial for NPC progression through ceRNA. In this article, the ceRNA regulatory network in NPC was established. It has been verified that AC008083.2, as the ceRNA of miR-142-3p, regulates NPC malignant progression, which greatly enriches the ceRNA regulatory network of NPC.

Cuproptosis is a type of cell death that occurs when copper ions accumulate in cells, resulting in oxidative stress and damage to cellular components. This process has been shown to have anti-tumor effects, as it can selectively kill cancer cells. In addition, cuproptosis has been studied as a potential treatment for NPC. The drug arsenic trioxide, which can induce cuproptosis, has a radiosensitizing effect on NPC cells in mice (*Chen et al., 2020*). Another study has investigated the role of miR-205-5p, a microRNA that regulates copper metabolism, in inducing cuproptosis in NPC cells. The study suggests that downregulating miR-205-5p can enhance the apoptotic effect of 3-bromopyruvate, a compound that induces cuproptosis, on NPC cells (*Shi et al., 2019*). These findings suggest that cuproptosis may have the potential as a therapeutic strategy for NPC. Therefore, cuproptosis might be a hopeful target for NPC therapy. According to 259 cuproptosis-related genes, the NPC samples were uniformly clustered. Based on different degrees of cuproptosis, the patients of NPC were classified into two groups. Furthermore, the differential expression analysis was conducted to get the genes related to cuproptosis in NPC. This article provides potential targets for the treatment of NPC with cuproptosis.

STRN3 is a gene that encodes STRN3, a protein belonging to the striatin family. These proteins are regarded as regulatory parts of the protein phosphatase PP2A. STRN3 is linked to STRN-interacting proteins (STRIP1/2) and members of the germinal center kinase III family, including MAP4K4 (*Migliavacca et al., 2022*). STRN3 is detected in multiple tissues, including the brain, heart, and liver. Although the exact role of STRN3 has not been fully elucidated, it is believed to participate in calcium-dependent signaling, scaffolding, and cell proliferation. Furthermore, STRN3 has been shown to enhance cell proliferation and facilitate tissue invasion in tumors. For example, a recent study found that STRN3 cooperates with MAP4K4 to promote growth and tissue invasion in breast cancer cells. Additionally, STRN3 is involved in developing several human diseases, including cancer. For NPC, STRN3 may participate in tumor proliferation. For example, STRN3 is up-regulated in NPC tissues, and the knockdown of STRN3 inhibits NPC cell proliferation and migration (*Verbinnen et al., 2021*). The mechanism by which STRN3 promotes NPC proliferation is not fully understood, but it may be involved in regulating the cell cycle and activating signaling pathways. Based on these discoveries, it is indicated that STRN3 may be a promising therapeutic target for the treatment of NPC. The relationship between STRN3 and ceRNA needs to be further established. However, ceRNAs can control gene expression in tumors, including NPC, and STRN3 can promote cell proliferation in NPC. STRN3 may function as a ceRNA to regulate gene expression in NPC, but the exact mechanism needs to be further investigated. Some studies have identified specific ceRNAs that play a role in NPC progression, such as hsa_circ_0046263, which promotes NPC progression by up-regulating IGFBP3 expression (*Yin et al., 2020*). These studies established a co-regulatory network of STRN3 and non-coding RNAs in NPC, providing a theoretical basis for targeted STRN3 diagnosis and therapy.

## CONCLUSION

This research establishes that AC008083.2 functions as a competitive regulatory mediator for miR-142-3p, thereby competitively regulating the STRN3 expression associated with Nasopharyngeal Carcinoma (NPC). Moreover, the AC008083.2/miR-142-3p axis may play a crucial role in driving the progression of NPC through its influence on STRN3. Importantly, it is worth noting that AC008083.2's regulation of STRN3 expression and its impact on the malignant advancement of NPC partially relies on miR-142-3p. This investigation uncovers novel competing endogenous RNA (ceRNA) regulatory networks within NPC and unveils potential novel targets for the diagnosis and treatment of this disease.

## ACKNOWLEDGEMENTS

The authors thank the contributions of all scientists in the clinical and basic medicine field.

### Funding

The authors received no funding for this work.

### Competing Interests

The authors declare there are no competing interests.

### Author Contributions

- Dandan Feng conceived and designed the experiments, performed the experiments, analyzed the data, prepared figures and/or tables, authored or reviewed drafts of the article, and approved the final draft.
- Xiaoping Wu performed the experiments, analyzed the data, prepared figures and/or tables, and approved the final draft.
- Genping Li performed the experiments, prepared figures and/or tables, and approved the final draft.
- Junhui Yang performed the experiments, prepared figures and/or tables, and approved the final draft.
- Jianguo Jiang analyzed the data, prepared figures and/or tables, and approved the final draft.
- Shunan Liu analyzed the data, prepared figures and/or tables, and approved the final draft.
- Jichuan Chen conceived and designed the experiments, authored or reviewed drafts of the article, and approved the final draft.

### Data Availability

The datasets analyzed during the current study are available in the Cancer Genome Atlas: TCGA-HNSC, phs000178.

The raw data are available in the Supplemental File.

### Supplemental Information

Supplemental information for this article can be found online at http://dx.doi.org/10.7717/peerj.17859#supplemental-information.

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
