# Peer review of "Cuproptosis related ceRNA axis AC008083.2/miR-142-3p promotes the malignant progression of nasopharyngeal carcinoma through STRN3"

_PeerJ, doi:10.7717/peerj.17859_

## Round 0.1 · original submission · Major Revisions

Please address the concerns raised by reviewers.

Reviewer 1 ·

Basic reporting

In this manuscript, Feng, Chen and their colleagues investigated that AC008083.2 / miR-142-3p axis regulates malignant progression of NPC through STRN3 signaling. By performing bioinformatics analysis and in vitro cell assays, they observed that the changes in the expressions of AC008083.2/miR-142-3p/STRN3 can promote the proliferation of NPC. Moreover, mechanism studies also showed that the effects of STRN3 and AC008083.2 on NPC malignant progression depended on miR-142-3p. This study reveals an innovative ceRNA regulatory network in nasopharyngeal carcinoma. It is considered as a new potential target for diagnosis and treatment of nasopharyngeal carcinoma. These findings have important potential implications, and the study demonstrates a well-designed approach. I have the following suggestions.
1.It is recommended to change the image layout in the results section. For example, in picture 5, some parts don't seem to be aligned with it and it doesn't look very harmonious.
2.The English grammar needs to be further improvement.

Experimental design

no comment

Validity of the findings

The key molecules involved in this study, AC008083.2, miR-142-3p, and STRN3, have not been validated in tumor tissues.
The modes of regulation and the sites of action among the three main molecules require further experimental validation.

Additional comments

Major revisions

·

Basic reporting

In the manuscript entitled “"Cuproptosis Related CeRNA Axis AC008083.2/miR-142-3p Promotes the Malignant Progression of Nasopharyngeal Carcinoma through STRN3", Feng et al. investigate the AC008083.2/miR-142-3p/ STRN3 axis in NPC. Using bioinformatics and dual-luciferase reporter assays, the researchers demonstrate that the AC008083.2/miR-142-3p axis influences the malignant progression of NPC by regulating STRN3 expression. They discover that AC008083.2 can alter NPC cell proliferation by acting as a ceRNA for miR-142-3p, affecting STRN3 levels. Their findings highlight an ceRNA regulatory network, suggesting new diagnostic and treatment targets for NPC, and underscore the complexity of ncRNA-mediated gene regulation in cancer. The manuscript would benefit substantially from thorough revisions in both substance and style. The narrative is often overly simplistic, leading to potential misinterpretations. There are sections with conflicting statements and possible errors. A deep and comprehensive understanding of their research is vital for the authors, as it directly influences their ability to present their findings accurately and reliably. Given the substantial issues identified in this manuscript—perhaps more accurately described as a draft at this stage—I will outline some of the key points of concern and comments.
1.The authors have suggested that STRN3 is a key oncogene in nasopharyngeal carcinoma, but it's important to note that current research does not directly support STRN3's role as an oncogene in NPC. Therefore, this claim should be rephrased to accurately reflect the extent of evidence available, underlining the necessity for further studies to fully understand STRN3's involvement in the pathology of NPC.
2.Research into cuproptosis and its potential therapeutic applications is still in the early stages. Several studies have explored the mechanisms by which cuproptosis can be induced and its effects on cancer cells. For example, copper ionophores like elesclomol and disulfiram have shown potential in transporting copper ions into cells and mitochondria, inducing cuproptosis by causing aggregation of lipoylated proteins and destabilization of Fe-S cluster proteins (PMID: 37786749). However, most of these studies are primarily preclinical, focusing on cell lines, so the accuracy of the statement "Cuproptosis has important biological significance in tumor treatment" requires careful consideration.
3.In the line 37, It would be significantly beneficial if the author could include detailed clinical studies and statistical data to support the statement "The disease has a poor prognosis and high malignancy."
4.In the line 56, the term "anti-oncogenes" is less commonly used in current scientific literature, with "tumor suppressor genes" being the more prevalent term to describe genes that, when functioning normally, prevent the development of tumors by regulating cell growth and division.
5.In lines 57-58, where it's mentioned, "Understanding the miRNA-mediated mechanisms in NPC offers insights into its diagnosis, prognosis, and potential therapeutic targets," it would be beneficial for the authors to provide an overview of advancements in this field along with key discoveries.
6.The introduction of this draft could benefit from a clearer logical structure. It's recommended that the author reorganizes this section to ensure a smoother and more logic flow of ideas.
7.The method description in this draft is overly simplistic, making it difficult for readers to fully understand and replicate the work. Expanding this section with more detailed and precise explanations would greatly assist in clarifying the procedures used. For example, in the “Data collection” section, which kinds of data are downloaded from TCGA? RNA, DNA, methylation, or others? Additionally, information such as the version of the R package used, further insights into the bioinformatics analysis, and specific catalog numbers and vendors for reagents and kits used in this study, are also should be provided.
8.In the lines 148-152, the authors state that only one time point was detected, yet Figure 6B clearly shows three time points. This discrepancy calls for a clarification or correction in the draft to accurately reflect the data presented.
9.Is the differential expression analysis of lncRNAs and miRNAs, as shown in Figure 1, performed on tumor tissue compared to adjacent normal tissue? It should be clearly claimed in the figure legends and the results section.
10.How did the authors arrive at the conclusion regarding "up-regulated oncogenes and down-regulated suppressor genes" as stated in Lines 209-210? What methods or analyses were used to support this statement?
11.Could you clarify the discrepancy between stating "A total of 58 NPC samples were downloaded from the TCGA database" in line 126 and then mentioning "A total of 548 patients with NPC from TCGA..." in line 216? This inconsistency is quite confusing.
12.It's important to define abbreviations the first time they are used. For example, in line 218, when you mention "OS," could you specify that it stands for "Overall Survival"?
13.Why did the authors focus on the AC008083.2/miR-142-3p/STRN3 axis, given that miRNAs typically have multiple target genes, not just a one-to-one relationship? Could the authors provide the reasoning behind their research design?
14.The font size in some figures is too small for comfortable reading. Could they be made larger and consistent across all figures?
15.In the line 239, the authors believe that “The AUC curve shows a significant sensitivity”. however, the AUC values in the Figure 4C for the genes AC008083.2, AC078820.1, and AL160408.4 are 0.596, 0.519, and 0.53, respectively. These values are relatively close to 0.5, which is the AUC value of a random classifier, indicating that the predictive accuracy for these may not be strong. Therefore, describing these AUC curves as significantly sensitive could be misleading.

Experimental design

See the Basic reporting in detail

Validity of the findings

See the Basic reporting in detail

Additional comments

See the Basic reporting in detail

Reviewer 3 ·

Basic reporting

The manuscript by Feng and coworkers titled ‘Cuproptosis related ceRNA axis AC008083.2/miR-142-3p promotes the malignant progression of nasopharyngeal carcinoma through STRN3’ describes AC008083.2/miR-142-3p/STRN3 axis in NPC.

The manuscript started with bioinformatic analyses of NPC transcriptome data. This is followed by extensive invitro experiments to demonstrate the AC008083.2/miR-142-3p/STRN3 axis in NPC. Invitro data is comprehensive and rigorous. However, the bioinformatics portion is poorly presented and should be extensively revised.

1. The introduction is very lengthy. I suggest authors reduce the length of introduction. In Particular, paragraphs starting from line 50 and 75 can be shortened.
2. This manuscript requires extensive language editing. Many sentences are vague. I ask authors to check sentences like ‘….. valuable oncogene.’ and remove unnecessary words.
3. Abstract: The background paragraph in the abstract seems disconnected. I suggest explaining the lacuna and main objective clearly. Introduction on NPC, prognosis, lncRNas, miRNAs etc is not necessary. The main purpose of ‘background’ is to explain what is not known and what are the objectives.
4. Line 208: ‘ ……difference analysis ……’. Is it differential expression analysis.?
5. Lines 209-210: Please delete the terms ‘oncogenes’ and tumot suppressor gene’ in this sentence. Please use the following words instead: Upregulated genes and down regulated genes. I am suggesting this because all upregulated genes are not necessarily oncogenes and all down regulated genes are not necessarily TSGs.
6. Line 259: is it Pearson correlation?
7. Please label X axes in plots shown in figure 1B
8. What are group1 and 2 in figures 1C?
9. What is the rationale for consensus clustering shown in figure 2A? These results were poorly described and lacked clarity. Figure 2 seems to be very confusing and redundant. I suggest authors consider removing this figure. Figure 1 already shows prognosis-based on miRNAs and lncRNAs. Why was 1 out of 3 clusters excluded in further analysis? Can authors check their data again to confirm if there are 75% of the patients in G2 who died on day 5250? Lable X axis of figure 2E. Lines indicating 95% CI are not legible.
10. Legend of figure 4: Please write RNA levels instead of nucleic acid levels.
11. The AUROC values in figure 4C are very poor and indicate that the classifier prediction is just random (because AUROC of 0.5 indicates complete random prediction). Please remove this figure and corresponding text.
12. Figure 5A is not legible.

Experimental design

none

Validity of the findings

none

---

## Round 0.2 · accepted · Accept

All issues pointed by the reviewers were adequately addressed and the manuscript was amended accordingly. The revised manuscript is acceptable now.

Reviewer 1 ·

Basic reporting

The research question is well-defined and the study design is appropriate.

Experimental design

The data collection and analysis methods are robust.

Validity of the findings

The findings are presented in a clear and concise manner.

·

Basic reporting

Thanks for the author's reply, most of my comments have been addressed and the revise manuscript improved somehow, and now i have not any more additonal comments and concerns at this time.

Experimental design

Thanks for the author's reply, most of my comments have been addressed and the revise manuscript improved somehow, and now i have not any more additonal comments and concerns at this time.

Validity of the findings

Thanks for the author's reply, most of my comments have been addressed and the revise manuscript improved somehow, and now i have not any more additonal comments and concerns at this time.

Reviewer 3 ·

Basic reporting

Authors addressed my concerns.

Experimental design

-

Validity of the findings

-